# Anti-Inflammatory Action of Heterogeneous Nuclear Ribonucleoprotein A2/B1 in Patients with Autoimmune Endocrine Disorders

**DOI:** 10.3390/jcm9010009

**Published:** 2019-12-19

**Authors:** Antonina Coppola, Patrizia Cancemi, Laura Tomasello, Valentina Guarnotta, Maria Pitrone, Valentina Failla, Salvatore Cillino, Salvatore Feo, Giuseppe Pizzolanti, Carla Giordano

**Affiliations:** 1Laboratory of Regenerative Medicine, Section of Endocrinology, Diabetology and Metabolism, Department of Health Promotion, Mother and Child Care, Internal Medicine and Medical Specialties (ProMISE), University of Palermo, 90127 Palermo, Italy; antonina.coppola02@unipa.it (A.C.); laura.tomasello@unipa.it (L.T.); valentina.guarnotta@unipa.it (V.G.); maria.pitrone@unipa.it (M.P.); 2Laboratory of Cell Colture, ATeN (Advanced Technologies Network) Center, University of Palermo, 90128 Palermo, Italy; patrizia.cancemi@unipa.it (P.C.);; 3Department of Biological Chemical and Pharmaceutical Sciences and Technologies (STEBICEF), University of Palermo, 90128 Palermo, Italy; 4Department of Ophthalmology, University of Palermo, 90127 Palermo, Italy; vale.failla@hotmail.it (V.F.); salvatore.cillino@unipa.it (S.C.)

**Keywords:** immunotolerance, hnRNP A2/B1, autoimmunity, fibroblast-limbal stem cells, Autoimmune Endocrine Diseases, NF-ĸB interaction

## Abstract

Our previous studies documented that human fibroblast-limbal stem cells (f-LSCs) possess immunosuppressive capabilities, playing a role in regulating T-cell activity. This study highlights the molecular activities by which human f-LSCs can attenuate the inflammatory responses of self-reactive peripheral blood mononuclear cells (PBMCs) collected from patients with autoimmune endocrine diseases (AEDs). Anti-CD3 activated PBMCs from twenty healthy donors and fifty-two patients with AEDs were cocultured on f-LSC monolayer. 2D-DIGE proteomic experiments, mass spectrometry sequencing and functional in vitro assays were assessed in cocultured PBMCs. We identified the downmodulation of several human heterogeneous nuclear ribonucleoprotein A2/B1 (hnRNP A2/B1) isoforms in healthy and AED activated PBMCs upon f-LSC interaction. The reduction of hnRNPA2/B1 protein expression largely affected the cycling ki67^+^, CD25^+^, PD-1^+^ reactive cells and the double marked CD8^+^/hnRNPA2B1^+^ T cell subset. Anti-PD1 blocking experiments evoked hnRNPA2/B1 overexpression, attributing putative activation function to the protein. hnRNPA2/B2 transient silencing inverted immunopolarization of the self-reactive PBMCs from AEDs toward a M2/Th2-type background. Pharmacological inhibition and co-immunoprecipitation experiments demonstrated the involvement of NF-ĸB in hnRNPA2/B activity and turnover. Our data indicate cardinal involvement of hnRNP A2/B1 protein in peripheral mechanisms of tolerance restoration and attenuation of inflammation, identifying a novel immunoplayer potentially targetable in all AEDs.

## 1. Introduction

Autoimmune endocrine diseases (AEDs) are widespread disorders that utilize immense healthcare resources and public costs due to their permanent therapeutic monitoring. Autoimmune attack towards the endocrine system is their pathogenic basis. AEDs include type 1 diabetes mellitus, Hashimoto’s thyroiditis, Addison’s disease and polyglandular autoimmune syndromes (APS). In all these cases, the immune response to the target cell progressively destroys the endocrine gland, and the clinical manifestations are primarily the result of glandular hypofunction leading to inadequate hormone production. AEDs become more clinically insidious when the single endocrine diseases combine with each other in APS. They can be categorized as rare monogenic forms, such as APS-1, and a more common polygenic variety, APS-2 [1]. Although the gold standard of treatment for these disorders remains the replacement of hormones produced by the damaged endocrine organs, further investigations are required in order to develop novel specific therapeutic measures able to treat the immunological cause before their clinical manifestation.

Human Heterogeneous Nuclear Ribonucleoprotein A2/B1 isoforms are among the most abundant pre-mRNA-binding proteins of vertebrates, ubiquitously participating in RNA-binding and pre-RNA processing. They are constitutively expressed in lymphoid organs and highly upregulated in the synovial tissue of patients with rheumatoid arthritis (RA), who may also generate autoantibodies to this protein [2]. Mutations in the genes encoding hnRNPA2/B1 have been reported in a multi-system proteinopathy that includes amyotrophic lateral sclerosis, Paget’s disease and frontotemporal dementia [3]. Many of the hnRNPs have been directly or indirectly implicated in cancer development, including anaplastic thyroid cancer [4,5,6]. It is important to note that various hnRNPs contribute to disease progression via distinct mechanisms and exert their effects via pathways involving different sets of genes. No data have been reported about their possible involvement in AEDs.

Due to their regenerative and immunosuppressive properties, Mesenchymal Stem Cells (MSCs) derived from different adult tissue have been demonstrated to contribute to control of inflammatory diseases, maintaining intact their stem cell properties [7,8,9]. MSCs suppress T-, B- and dendritic cell function, providing a rational basis for their application in the treatment of numerous immune-mediated disorders [10]. The limbus is a highly specialized region of the eye hosting a well-recognized population of epithelial stem cells (LESCs), which continuously renew the corneal surface [11]. There is more recent evidence that the limbal niche also hosts a stromal fibroblast-like stem cell (f-LSC) population. We recently demonstrated that f-LSCs possess multilineage transdifferentiation potential and fail to induce a full allogeneic T-lymphocyte response, showing immunomodulatory capabilities [12,13]. These cells lack HLA-DR and produce cell contact/soluble factors able to maintain their immune privileged status even with an inflammatory background. When treated with Th1 cytokines (IL-6, IL-1β, IFN-γ), f-LSCs improve their phenotype and their expression for several immune regulatory mediators, acting as “smart immunomodulators.” Hence their immunosuppressive potential is expected to be superior in all autoimmunity disorders, including AEDs. We previously found that the therapeutic effects of f-LSCs are associated with downregulation of Th1-driven inflammatory responses in peripheral blood mononuclear cells (PBMCs) of female patients with Hashimoto’s thyroiditis (HT) after coculture. In the present study, we aimed to discover, through a proteomic approach, whether there exists a principle molecular player responsible for f-LSC-induced immunoeducation.

## 2. Materials and Methods

### 2.1. Isolation of Limbal Stem Cells

Human corneo-scleral rings from donors were processed as previously described [12]. The study was approved by the Ethical Committee of the AOUP, University of Palermo (No. 09/2009).

Briefly, samples underwent fine dissection with a sterile blade and were subsequently incubated with collagenase I (5 mg/mL; Sigma-Aldrich, St. Louis, MO, USA) overnight at 37 °C in a shaking bath and the day after, placed in p60 culture dishes (Corning, New York, USA) with the fibroblast maintenance medium (DMEM/F12 supplemented with 10% embryonic stem cell-tested foetal bovine serum (EC-FBS; PAA Laboratories Gmbh, Austria), 1× ITS (5μg/mL insulin, 5μg/mL transferrin, 5μg/Ml selenium; PAA Laboratories) and 20 ng/mL basic fibroblast growth factor (b-FGF; Preprotech, London, UK) until the cells reached confluence. The f-LSC subcultures were kept in the expansion medium (DMEM/F12 supplemented with 5% embryonic stem cell-tested foetal bovine serum (EC-FBS; PAA Laboratories), 1× insulin-transferrin-selenium (ITS; PAA Laboratories) and 4 ng/mL basic fibroblast growth factor (Preprotech) up to passage 20.

### 2.2. Patient Selection

The study was carried out in accordance with the recommendations of the Paolo Giaccone Policlinico Ethics Committee and approved by the Ethical Committee (No. 07/2018). Written informed consent was obtained from all subjects, in accordance with the Declaration of Helsinki. Eligible subjects were females and males who had been diagnosed within at least 12 months. Thirty-five female patients, aged between 28 and 66, with autoimmune thyroiditis and elevated plasma TPOAb and/or Tg antibodies (TgAb) were selected at the Outpatient Clinic of the Section of Endocrinology, P. Giaccone Policlinico, University of Palermo, Italy. All patients were receiving L-T4 replacement therapy at an appropriate dosage to maintain basal TSH within the normal range. Blood samples were also collected from 17 patients with primary adrenal insufficiency (PAI). All these patients were on stable glucocorticoid replacement therapy before entering the study and were consecutively referred to the Division of Endocrinology of Palermo University from January 2016 to December 2017. Fourteen out of the 17 enrolled patients were females, mean age 55 years old, with isolated PAI (nine patients) or APS (five patients). The five patients with APS included one patient with combined celiac disease and PAI, three patients with combined type 1 diabetes mellitus, hypothyroidism and PAI and only one carrying AIRE mutations that confirmed the clinical diagnosis for APS-1 [14]. Three out of the 17 patients were males, mean age 50 years old, with isolated PAI. The healthy control group comprised 20 subjects, of which 13 males and 7 females aged 24–34 years. Fifteen milliliters of heparin anticoagulated blood was drawn from each donor/patient in the morning after a 12-hour fasting period. The blood was diluted 1:1 with PBS solution, and PBMCs were separated by gradient centrifugation over Ficoll (Lympholyte-Human Cell Separation Media, Cedarlane, Burlington, Canada) according to the manufacturer’s instructions.

### 2.3. Cell Culture and Cocultivation Experiments

PBMCs obtained by means of a Ficoll-Paque density gradient were cultured in complete RPMI-1640 (PAA Laboratories) supplemented with 10% heat-inactivated foetal bovine serum (BSA; PAA Laboratories). In coculture experiments, the PBMCs were activated for 72 hours with 5 μg/mL of mAb anti-human CD3 (OKT-3 Clone, SigmAldrich, Milan, Italy). L-FSCs were seeded at 2000 cells/cm^2^ and allowed to adhere in 24-well or 6-well plates (Corning Co, Corning, NY) overnight. After 24 hours, they were cocultured with total PBMCs at a 1:50 ratio and gently recovered as the supernatant as previously reported [13]. Conditioned medium for each cell culture was dialyzed for 24 h under agitation against H2O milliQ and then lyophilized. PBMCs without TCR-stimulation were used as negative controls.

Jurkat T leukemia cells were cultured in RPMI 1640 medium (GIBCO laboratories, Milan, Italy) supplemented with 10% FCS, activated for 5 hours with 10µg/mL of Phytohaemagglutinin (PHA, Sigma Aldrich, Milan, Italy) at 37 °C and 5% CO_2_.

### 2.4. Proteomic Analysis

f-LSCs, PBMCs+antiCD3 and PBMCs+antiCD3 cocultured with f-LSCs were lysated with Lysis buffer (7 M urea, 2 mM Thiourea, 0.4 % *w*/*v* CHAPS, 1 % *w*/*v* 1,4-dithioerythritol (DTE) and 30 mM TRIZMA base, pH 8.5). Conditioned media were prepared as previously described [15]. Protein concentration was determined by Bradford assay.

### 2.5. 2D-DIGE Analysis

Protein samples (50 μg) were labeled for 2D-DIGE analysis using a CyDyeTM DIGE minimal labeling kit (GE Healthcare, Gothenburg, Sweden), [16]. The first-dimension separation was performed at 20 °C on commercial sigmoidal immobilized pH gradient strips (IPG), 18 cm long with pH range 3.5 to 10 (Pharmacia), as previously described [17]. The focused proteins were then separated on 9%–16% linear gradient polyacrylamide gels (SDS-PAGE) using a DALT six (GE Healthcare) apparatus with a constant current of 40 mA/gel at 10 °C. Images were acquired with a Typhoon FLA 9500 scanner (GE Healthcare), using specific emission filters, and analysed by the *Image* Master 2D Platinum 7 software (GE Healthcare). A total of six gels was run to achieve a statistically significant measure of the differences in protein expression. Protein spots showing more than a 1.3-fold change in spot volume (increased for up-regulation or decreased for down-regulation), with a statistically significant ANOVA value (*p* ≤ 0.05), were considered differentially represented and further identified by MS analysis. After acquisition, each gel was stained with ammoniacal silver nitrate.

### 2.6. In-gel Digestion and MS Aanalysis of Tryptic Digests

Spots of interest were manually picked and mass spectrometric sequencing was carried out after in-gel digestion, using sequencing-grade trypsin (20 μg/vial), as previously described [18]. The tryptic peptide extracts were dried in a vacuum centrifuge and dissolved in 0.1% trifluoroacetic acid (TFA). Peptide mixtures were desalted by μZip-TipC18 (Millipore, Milan, Italy). The matrix, R-cyano-4-hydroxycinnamic acid (HCCA), was used as a saturated solution of HCCA (1 μL) at 10 mg/mL in CH3CN/H2O (50:50 (v/v)) containing 0.1% TFA. Mass spectra were obtained using an Ultraflex MALDI-TOF-TOF (Bruker Daltonics, Bremen, Germany) mass spectrometer. Peptide mass fingerprinting was compared to the theoretical masses from the Swiss-Prot or NCBI sequence databases using Mascot (http://www.matrixscience.com/). Typical search parameters were as follows: 50 ppm of mass tolerance, carbamidomethylation of cysteine residues, one missed enzymatic cleavage for trypsin; a minimum of four peptide mass hits was required for a match; methionine residues could be considered in oxidized form; no restriction was placed on the isoelectric point of the protein; and a protein mass range from 5 to 100 kDa was allowed.

### 2.7. Western Blot

Protein samples (90 μg) were subjected to 2D-IPG gel electrophoresis and then electrotransferred into a nitrocellulose membrane [19]. Western blotting analyses were performed using a mouse monoclonal antibody for hnRNPA2/B1 (1:2000) diluted in 1% milk ON at 4 °C. Following incubation with the anti-mouse peroxidase-linked antibody (1:3000) for 1 h at room temperature, the reaction was revealed by the ECL detection system. The correct protein loading was ascertained by red Ponceau staining and immunoblotting for ACTB.

### 2.8. Isolation of Total RNA and qRT-PCR

Total RNA was extracted and purified from PBMCs using the RNeasy Micro Kit (Qiagen, Milan, Italy), according to the manufacturer’s protocol. An amount of 1 μg total RNA were reverse transcribed in a volume of 20 µl with Oligo dT primers (Applied Biosystems, Darmstad, Germany) and Stratascript RT (Stratagene, Amsterdam, Netherland). The primer pair sequences are listed in Appendix A. PCR primers for hnRNPA2/B1, IL-6, FAS, IDO, MCP1, CCND1 and p27 were purchased from Qiagen (QuantiTect Primer Assays, Qiagen, Milan, Italy). All reactions were performed with Quantitect Sybr Green PCR Kit (Qiagen) using the Rotor-Gene Q instrument (Qiagen, Milan, Italy) as previously described [20]. The specificity of the amplified products was determined by means of melting peak analysis. Relative gene expression analysis for each gene was performed with Rotor-Gene Q software using the Delta Delta Ct method validated according to the guidelines of Livak and Schmittgen [21]. All reactions were performed at least in triplicate.

### 2.9. Flow Cytometry

The cells were treated with an FcR blocking reagent (Miltenyi Biotec, Bergisch Gladbach, Germany) and incubated with each fluorochrome-conjugated antibody or appropriate isotype control at 4 °C for 30 minutes in the dark. For hnRNPA27B1 staining, cells were fixed for 15 minutes at 4 °C with 2% paraformaldehyde (PFA) and washed with staining buffer. The T-cell phenotype was determined using CD25 PerCP-Cy™5.5, PD-1 (CD279) PE, CD4 FITC, CD8 PE all purchased from BD Biosciences. Intracellular staining for hnRNPA2/B1 (Abcam, cat. no. ab6102; dilution 1:200, Milan, Italy) and Ki67 PE (BD, Bioscience) was performed using BD Cytofix/Cytoperm™ Plus Fixation/Permeabilization Kit (BD Biosciences, Milan, Italy) according to the manufacturer’s instructions. Secondary antibodies were incubated for 1 h at 4 °C (Alexa Flour 488- or PE-conjugated anti mouse IgG was used at dilution 1:300, Thermo Fisher Scientific, Milan Italy) and samples washed twice before reading. All data were acquired on a FACSCalibur and analysed using CELLQuest Pro software (BD Pharmingen, San Jose, CA). Compensation controls for each fluorochrome with positive and negative populations, unstained and IgG controls were acquired using the BD Calibrite 3-Color Kit Beads (Catalog n° 340486, BD, Milan, Italy) and single/double positive antibodies for the FL1, FL2 channels using fresh activated PBMCs.

### 2.10. Immunofluorescence

Cells were fixed for 15 min at room temperature (RT) in 2% (*w*/*v*) paraformaldehyde, permeabilized with 0.5% PBS/Saponin (Sigma-Aldrich), washed in PBS, and blocked for 30 min in 3% PBS/BSA (bovine serum albumin). Primary antibodies were incubated for 24 h at 4 °C, and secondary antibodies for 1 h at RT. Fluorescence microscopy measurements were performed using a Zeiss microscope (AXIO Vert. A1, Jena, Germany) and a 40× air objective and proper filters for excitation and emission lights. Fluorescence intensity quantification was performed using Image J software (version:1.52p).

### 2.11. PBMCs Electroporation and hnRNPA2/B1 siRNA Knockdown

Prior to electroporation, PBMCs were washed twice with serum-free RPMI, followed by an additional washing step with OptiMEM (Invitrogen) and resuspended to a final concentration of 2×106 PBMCs in 500 µL OptiMEM. Next, 50 nM/106 cells siRNA or fluorescent siRNA Control (catalog numbers: sc-43841 and sc-36869 Santa Cruz Biotechnology, Milan, Italy, respectively) was added to the cell suspension. Cells were electroporated in a 4 mm electroporation cuvette with a square-wave pulse (300 V and 50 ms) a GenePulser Xcell electroporation device (Bio-Rad, Milan, Italy). After electroporation, PBMCs were resuspended in RPMI supplemented with 10% of foetal bovine serum and anti-human CD3 mAbs. Cells were further incubated for 72 h. As a negative control of electroporation, No-targeting siRNA control was added to the cells before electroporation.

In f-LSCs, hnRNPA2/B1 was silenced using INTERFERin transfection agent (Polyplus Transfection, Milan, Italy), according to the manufacturer’s instructions. Briefly, cells were seeded into six-well plates at a density of 2.5 × 105 cells/well. The transfection agent and siRNA (100 nM) complex were added to the cells and incubated for 72 hours before mRNA collection.

### 2.12. In vitro Pharmacological Treatments

For the PD-1 blocking assay, anti-PD-1 human recombinant protein (Purified NA/LE Mouse anti-Human CD279, Clone EH12.1, BD Biosciences, Milan, Italy) was added every day for a final concentration of 20 μg/mL in 24-well culture plates. PBMCs were cultured for 3 days together activating anti-CD3 mAbs and harvested for qRT-PCR assays. Cells cultured with anti-CD3 alone or after adding f-LSCs were used as controls.

To induce or block expression of the NF-ĸB target genes, 5µM of Phytohemagglutinin (PHA) or 10 µM of parthenolide were respectively added in Jurkat T cells for 5 hours. Untreated cells were used as negative controls. Cells were harvested for qRT-PCR analysis using IL-2 mRNA expression levels as a positive control gene downstream NF-ĸB signalling.

### 2.13. Co-Immunoprecipitation Assay (Co-IP)

Jurkat cells were lysed in RIPA buffer supplemented with protease inhibitor cocktail (Roche), immunoprecipitated with 2 μg of hnRNPA2/B1 antibody (Abcam, ab6102) or control rabbit IgG antibodies (Sigma-Aldrich, St. Louis, USA) overnight at 4 degrees. Immuno complexes was captured by protein G beads (sc-2003, Santa Cruz Biotechnology, Milan, Italy) and washed 4 times with lysis buffer and heated in 40 µl of 1× loading buffer. An amount of 20 µl of samples were analysed by SDS-PAGE by blotting with primary (NF-ĸB p65 (c-20), sc-372 rabbit polyclonal antibody, Santa Cruz Biotechnology or hnRNPA2/B1 mouse monoclonal antibody, Abcam, cat. no. ab6102; dilution 1:200, Milan, Italy) and respective secondary antibody.

### 2.14. Statistical Analysis

Quantifications and statistical analyses in imaging acquisitions and in Western blotting were performed using ImageJ and GraphPad Prism 7. Quantifications and statistical analyses in FACS analyses were performed in CellQuest. The data are expressed as means ± SD and compared using the appropriate version of Student’s unpaired or paired t test. Test results are reported as one or two-tailed *p* values; *p* < 0.05 was considered statistically significant.

### 2.15. FACS Analyses Were Performed in CellQuest

The data are expressed as means ±SD and compared using the appropriate version of Student’s unpaired or paired *t*-test. Test results are reported as one or two-tailed *p* values; *p* < 0.05 was considered statistically significant.

## 3. Results

### 3.1. Proteome and Secretome Analysis

Differential proteomic and secretome analysis (2D-DIGE) was performed to detect protein modulations induced by f-LSCs in cocultured PBMCs. Figure 1A shows a prototype of proteomic maps of f-LSCs, activated PBMCs from healthy donors and coculture of both cell lysates and conditioned media. A total of 62 protein spots were identified by MALDI-TOF spectrometry. In particular, 42 protein spots were identified in the cell lysates, and 20 protein spots in the conditioned media. The predicted protein–protein interaction networks and a complete list of the intracellular and secreted modulated proteins are reported in Appendix A, respectively. Among the differentially expressed proteins, we focused our attention on different isoforms of hnRNPA2/B1 protein, detected in multiple spots (three isoforms in the cell lysates and two isoforms in the secretome). This is because they were differentially modulated in both cell lysates and secretome. The panel in Figure 1A represents a magnification corresponding to the area in which the identified hnRNPA2/B1 spots are focused in both cell lysates and conditioned media. The different isoforms were labeled with different alphabetic letters starting from the more acidic one. Quantitative analysis media showed significant downmodulation of the hnRNPA2/B1 “c” isoform (1.12 ± 0.08 vs. 0.90 ± 0.06), while the other two (hnRNPA2/B1 “a” and “b”) were found to be significantly upregulated in the secretome analysis (0.031 ± 0.02 vs. 0.058 ± 0.03 respectively), (Figure 1B). In order to normalize the total amount of the secreted mediators in the cultured media we calculated the sum of the proteins individually released by f-LSCs and PBMCs in the presence of anti-CD3 mAbs divided by two, as already reported by Burrows et al., 2015 [22]. The combination of the increased hnRNPA2/B1 secretion coupled with its decreased intracellular levels would imply dramatic attenuation of the biological activity of the protein in immunoeducated PBMCs.

### 3.2. Downmodulation of hnRNPA2/B1 Expression in AED Patients

The qRT-PCR analysis showed a significant decrease of hnRNPA2/B1 mRNA expression in cocultured and activated PBMCs of patients compared to the activated PBMCs alone (2.31 ± 0.86 vs. 1.24 ± 0.20, 1.28 ± 0.27 vs. 0.47 ± 0.37 and 1.36 ± 0.30 vs. 0.18 ± 0.74 in thyroiditis, Addison’s disease and APS respectively) (Figure 2A). Significantly, the inhibition of hnRNPA2/B1 mRNA was higher in PBMCs collected from patients than in those from healthy controls (53 ± 6.1% vs. 32 ± 5.0% respectively). The same trend was confirmed at the protein level by assessing immunofluorescence analysis in AED cocultured PBMCs for 72 h (Figure 2B).

Intensity quantification with ImageJ software revealed a 40% ± 4.8 reduction of fluorescence intensity (1.19 ± 0.22 vs. 0.55 ± 0.08) in cocultured PBMCs compared to anti-CD3 activated PBMCs. Significantly, several hnRNPA2/B1-positive PBMCs disappeared when cocultured with f-LSCs, while in merge pictures the DAPI positive cells persisted. Untreated PBMCs served as negative controls.

1D-western blot (Appendix A) and 2D-IPG western blot (Figure 3A) were also assessed. The bidimensional loading of samples from AED patients revealed our previous hnRNPA2/B1 isoforms identified in healthy controls (isoforms a, b and c) and three other different isoforms, referred to as “d”, “e” and “f”. The relative hnRNPA2/B1 expression (Figure 3B) showed more marked downmodulation of isoforms “a” and “b”, while isoforms “c” and “f” were totally absent in the f-LSC treated samples compared to anti-CD3 activated PBMCs. The expression levels of hnRNPA2/B1 “d” and “e” remained unchanged after cocolture. This phenomenon could be due to a higher inflammatory background level present in the T-cells collected from patients that consequently ameliorated the f-LSCs performance [23]. These data suggest a hypothetical correlation between hnRNPA2/B1 downregulation with self-reactive phenotype attenuation in PBMCs of AED patients and possible involvement of hnRNPA2/B1 protein in the pathophysiology of AEDs.

### 3.3. hnRNPA2/B1 Downmodulation Affects Proliferating and Activates Lympho-Monocytes of AED Patients

Compared to PBMCs alone, in activated PBMCs of AED patients the percentage of hnRNPA2/B1^+^Ki67^+^ increased from 1.2% ± 0.5 to 15.6% ± 1.3, confirming their proliferating status (Figure 4A). After f-LSC coculture, hnRNPA2/B1+Ki67+ PBMCs were reduced by 41% ± 4.5. The hnRNPA2/B1^+^: hnRNPA2/B1^−^ ratio was significantly higher in activated PBMCs compared to untreated cells and significantly lower after f-LSCs coculture (1.5 ± 0.2 vs. 0.14 ± 0.05 and 0.38 ± 0.05, respectively), (Figure 4B). This analysis reveals that the total decrease of hnRNPA2/B1 largely affected the proliferating portion of cells after immunomodulation induction.

In our system, all CD25+ and PD-1+ cells in activated PBMCs were found to be simultaneously positive for hnRNPA2/B1 (22.5% ± 3.5; 4%; ± 0.5 respectively), suggesting possible involvement of the hnRNPA2/B1 protein in the lymphocyte activation process (Figure 4C). Again, attenuation of the reactive phenotype in activated PBMCs after f-LSC treatment significantly reduced the hnRNPA2/B1^+^: hnRNPA2/B1^−^ ratio from 13.3 ± 2.3 to 5.9 ± 0.5 and from 10.8 ± 0.5 to 7.3 ± 1.6 in CD25/hnRNPA2/B1 and PD-1/hnRNPA2/B1 co-stained cells, respectively (Figure 4D,E). The hnRNPA2/B1^+^: hnRNPA2/B1^−^ ratio in untreated cells was used as a negative control in CD25 and PD-1 detection (4.3 ± 1.0 and 3.2 ± 0.7 respectively). These findings suggest a novel putative activation function for the hnRNPA2/B1 protein, acting in the same way as other conventional T-cell activation markers.

### 3.4. hnRNPA2/B1 Displays Immunoregulation Function Mainly on the CD8+ T Cell Subset

Flow cytometry data revealed that hnRNPA2/B1 was constitutively expressed in CD4+, CD8+ and CD14+ cells in a percentage of 37% ± 6.2, 27% ± 5.5 and 14% ± 2.5, respectively, in samples of AED patients (Appendix A). To explore the possible regulatory effect of f-LSCs on CD4+ or CD8+ T subsets, flow cytometry analysis of anti-CD4^+^ and/or anti-CD8^+^ in hnRNPA2/B1^+^ was performed. After PBMC activation, the hnRNPA2/B1^+^: hnRNPA2/B1^−^ ratio in hnRNPA2/B1-CD4 or hnRNPA2/B1-CD8 co-stained AED samples increased compared to untreated PBMCs (4.7 ± 1.1 vs. 32 ± 11 and 2.4 ± 0.8 vs. 13 ± 4.1 respectively). After coculture, the same ratio was appreciably downregulated on both CD4+ or CD8+ T cells, validating our previous data (32 ± 11 vs.18.5 ± 9.0 and 13 ± 4.1 vs. 6.2 ± 1.5 respectively), (Figure 4F, upper panels). Only the CD8+ hnRNPA2/B1+ subpopulation was found to be significantly decreased after f-LSC induction (35% ± 6.5 vs. 27% ± 5.1), (Figure 4F, lower panels), indicating that f-LSCs displayed immunosuppression mainly on the double positive CD8+ hnRNPA2B1+ T cell subset.

### 3.5. Anti-PD-1 Treatment Induces hnRNPA2/B1 Overexpression

In the presence of soluble anti-human PD-1 an up to two-fold increase in proliferative response was observed by manual cell counting. Similarly, increased levels of mRNA for IL-2, Bcl-2, IFN-γ, all survival markers downstream of the T cell receptor (TCR), were found (3.8 ± 1.0; 1.4 ± 0.1; 1.5 ± 0.1 respectively vs. 1 ± 0.1), (Figure 5A). As a consequence of the PD-1 blockade, the PDL-1 mRNA was significantly downmodulated compared to anti-CD3 activated PBMCs (0.86 ± 0.04 vs. 1 ± 0.09 respectively). Furthermore, the same cells exhibited constitutive levels of cyclin D1 but lower mRNA levels of the negative cell cycle regulatory protein p27Kip1 (0.48 ± 0.06 vs. 1 ± 0.09) probably to induce cell cycle entry. An inverse expression trend, in response to f-LSC treatment, was observed for the same genes (0.27 ± 0.08 for IL-2, 0.58 ± 0.02 for Bcl-2, 0.46 ± 0.04 for IFN-γ and 1.5 ± 0.08 for p27Kip1) suggesting the possibility that f-LSCs exerted their growth inhibitory effect through induction of p27Kip1 expression.

### 3.6. hnRNPA2/B1 Silencing Induces M2/Th2 Polarization in Self-Reactive PBMCs of AED Patients

At 72 h after electroporation, with harvesting of activated PBMCs for mRNA extraction we obtained silencing of 70 ± 8% compared to Non-targeting siRNA Control. No significant changes in expression for the Forkhead box P3 (FOXP3) transcription factor, TGF-β and interleukin 17A (IL-17A) were found. These data reveal the inability to expand the Tregs or to attenuate Th17 immune activation via hnRNPA2/B1 downregulation. Furthermore, Arginase-1 (Arg1) and interleukin 4 (IL-4), prototypic markers of M2 activation, were significantly upregulated after silencing (1.53 ± 0.05 vs. 1.0 ± 0.2 and 3.2 ± 0.4 vs. 1.0 ± 0.3 respectively). By contrast, the monocyte chemoattractant protein-1 (MCP-1/CCL2), a key chemokine able to recruit monocytes in foci of active inflammation, was found to be nine-fold downregulated (0.12 ± 0.05 vs. 1.0 ± 0.2) after siRNA transfection (Figure 5B).

### 3.7. HnRNPA2/B1 Knockdown Improves Immunosuppressive Potential of f-LSCs

To understand the importance of hnRNPA27B in tolerance induction, transfection-based studies in f-LSCs were performed. The immunosuppressive action of MSCs was reported to be mediated by production of both cell contact and soluble factors [24,25]. Among them, after hnRNPA2/B1 silencing in f-LSCs (84% ± 4.8 of transfection efficacy), Indoleamine-pyrrole 2,3-dioxygenase (IDO), Cyclooxygenase-2 (COX-2), hepatocyte growth factor (HGF), programmed death-ligand 1 (PDL-1/2) and interleukin-6 (IL-6) were all found to be increased in expression (3.9-, 4.6-, 1.3-, 5.1- and 1.5-fold respectively), (Figure 5C). No change in TGF-β and FAS expression was observed due to an independent role of the hnRNPA2/B1 protein in T regulatory expansion or apoptosis induction. Taken together, these findings demonstrate that the immunosuppressive effects of f-LSCs are accompanied by a cytokine expression pattern that is significantly hnRNPA2/B1-dependent.

### 3.8. hnRNPA2/B1 mRNA Levels Are Influenced by NF-ĸB Transcription Fctor Activity in Jurkat T Cells

As expected, the mRNA expression levels of IL-2 were highly induced by PHA treatment and reduced upon parthenolide treatment up to 19-fold. Under the same conditions, in the presence of parthenolide the hnRNPA2/B1 mRNA expression was downmodulated by 38% ± 1.1 and its level was 1.6-fold or 1.8-fold lower compared to the level before activation or after PHA activation respectively, (Figure 6A). Finally, Jurkat cells were lysed for protein extraction and hnRNPA2/B1 was immunoprecipitated using hnRNPA2/B1 antibody. Immunoblotting for p65 RelA confirmed the co-precipitation of hnRNPA2/B1 with p65/RelA component. This physical interaction suggests that possible involvement of the hnRNPA2/B1 protein in the NF-ĸB transcriptional function (Figure 6B).

## 4. Discussion

Identification of new targets specifically expressed in autoreactive immune cells could represent a possible step toward autoimmune disease resolution. Autoimmune disorders afflict 5%–10% of the population and a sizeable percentage of these pathologies involve an untoward immune response against endocrine organs [26]. In this connection, any endocrine organ, or more than one, can be targeted by the immune system as part of an autoimmune response, as in the case of APS. At the present, no clinical approach has aimed to treat the immunological cause of AEDs and it is only possible to control their symptoms and clinical manifestations through lifetime hormone replacement therapy. Alternative splicing of pre-mRNA is considered to be important to regulate hormonal activity. Heterogeneous nuclear ribonucleoproteins (hnRNP) A2 and B1 are two of the abundant nuclear RNA-binding proteins involved in thyroid hormonogenesis [4]. Here, for the first time, we present the possible involvement of the hnRNPA2/B1 protein in tolerance induction and attenuation of inflammatory immune responses.

In this study, to better define the f-LSC-mediated immunomodulatory mechanisms, we performed a 2D-DIGE proteomic analysis on anti-CD3 activated PBMCs of healthy controls and AED patients cultured in the presence or absence of f-LSCs. In particular, among the most representative modulated proteins, we identified three isoforms of hnRNPA2/B1 that we conventionally called “a”, “b” and “c”. More specifically, we discovered significant downmodulation of the intracellular isoform “c” and significant oversecretion of the other two isoforms “a” and “b” in activated PBMCs of young healthy donors cocultured with f-LSCs. This phenomenon suggests a considerable weakness of the biological function of the protein in immunoeducated PBMCs. In order to explore the possible involvement of hnRNPA2/B1 in AED physiopatology, we assessed qRT-PCR in f-LSC-treated PBMCs to detect the mRNA levels of our target gene. Interestingly, our data reveal more pronounced downmodulation of hnRNPA2/B1 mRNA in AED patients compared to healthy donors. This effect may be due to an inflammatory background typically detectable in an autoimmunity contest and able to stimulate the immune process in which hnRNPA2/B1 activity is likely involved. The same kind of trend was observed in immunofluorescence experiments, where we found a reduction in the total number of positive cells for hnRNPA2/B1 rather than a generic decrease in fluorescence intensity. The vanished hnRNPA2/B1^+^ PBMCs could correspond to those cells that acquired a tolerogenic phenotype after f-LSC treatment. We further separated the total cellular protein extracts using 2D-IPG gel electrophoresis. Three new protein spots that reacted with the hnRNPA2/B1 monoclonal antibody were detected. The “c” and “f” isoforms were found to be totally absent while the “a” and “b” isoforms were significantly reduced in f-LSC treated PBMCs of AED patients. The new expression balance observed could correspond to the portion of unexpressed protein found in healthy donor samples.

To explain the biological meaning of the hnRNPA2/B1 reduction in “immunoeducated” PBMCs, we performed a flow cytometric analysis for Ki67 and two conventional activation markers (CD25 and PD-1). It is well known that the hnRNPA2/B1 protein is involved in cell cycle progression, tumorigenesis and stemness [27,28,29]. The concomitant reduction of hnRNPA2/B1 and Ki67 in our coculture system suggests a cardinal function for the hnRNPA2/B1 protein in f-LSC-mediated T-cell unresponsiveness known as “tolerance arrest” [30].

Activation of lymphocytes is a complex cascade of events that results in the expression of cytokine receptors, the production and secretion of cytokines, and expression of several cell surface molecules that eventually lead to divergent immune responses [31].

Flow-cytometry-based measurements of the T-cell surface markers, CD25 and PD-1, were performed for monitoring the f-LSC induced immune changes. CD25 is the alpha chain of the trimeric IL-2 receptor and considered to be the most prominent cellular activation marker [32]. Programmed cell death 1 (PD-1) is a member of the CD28 family primarily upregulated on the surface of CD4 and CD8 T cells upon activation [33,34]. It also limits excessive immune responses to antigens and prevents autoimmunity [35]. Attenuation of the reactive phenotype in activated PBMCs after f-LSC treatment strongly affected the decreased double positive hnRNPA2/B1-CD25 or -PD-1 populations. These data suggest that the hnRNPA2/B1 could act as a fundamental immunoreactive protein, like other conventional T-cell activation markers. In addition, our flow cytomectric experiments always showed significant impairment of the hnRNPA2/B1^+^:hnRNPA2/B1^−^ ratios during the CD25, PD-1, CD4 and CD8 detection, improving the robustness of our data. To study the possible sharing of CD4+ or CD8+ T subsets in immunoregulation, we further performed staining for CD4 or CD8 and the hnRNPA2/B1 protein before and after f-LSC treatment. The moderate reduction in the percentage of CD8+ hnRNPA2/B1+ cells under f-LSC stimulation suggests that f-LSCs could display regulation on the activation *status* of CD8+ T-subset via hnRNPA2/B1 downregulation.

Previous studies postulated that PD-1 blockade might result in enhancement of T cell responses [36,37]. To test this hypothesis, we also assessed anti-PD1 treatments in the presence or absence of anti-CD3 mAbs, as an alternative in vitro strategy of T cell expansion/activation. As expected, the IL-2, Bcl-2 and IFN-γ mRNA levels were all found to be upregulated in AED PBMCs co-treated with anti-PD1 and anti-CD3 mAbs compared to anti-CD3 treated PBMCs. An inverse expression trend was observed for the cell cycle inhibitor p27kip. The increase in the proliferative and activation rate in cocultured PBMCs was accompanied by hnRNPA2/B1 mRNA induction, clarifying its possible role as an immunoactivator potentially implicated in the pathophysiology of AEDs. To better explore this aspect, efficient silencing of the gene was achieved in activated PBMCs of patients by electroporation. Macrophages and lymphocytes have broadly been characterized as either classically activated (M1/Th1) or alternatively activated (M2/Th2) based on surface receptors, gene signatures and secretion of inflammatory mediators [38,39]. It was reported that expression levels of the surface markers for human M1 and M2 macrophages were independent of granulocyte/macrophage colony-stimulating factor (GM-CSF) or macrophage colony-stimulating factor (M-CSF) treatment but dependent on the presence of Th1 or Th2 cytokines, underlining the importance of the cytokine environment for macrophage polarization [40,41]. In our knockdown experiments, the downmodulation of hnRNP A2/B1 in the autoreactive PBMCs did not interfere with production of the transcription factor FOXP3 and the two cytokines TGF-β and IL-17A, suggesting its inability to attenuate Th17 differentiation or to induce Treg expansion. By contrast, the IL-4 and Arginase-1 genes, as conventional Th2/M2 markers, were found to be upregulated. Anti-inflammatory polarization was also associated with a decrease in the inflammatory cytokine MCP-1. These findings define a crucial role of hnRNP A2/B1 in establishing the cytokine balance in PBMCs of AED patients.

Although it is indisputable that MSC therapy contributes to immunosuppression, further elucidations of the detailed biological mechanisms involved in this process are required [42,43,44]. To address the anti-inflammatory role of hnRNP A2/B1 in f-LSCs, we performed a small interfering RNA (siRNA)-mediated functional knockdown. After hnRNPA2/B1 silencing IDO, COX-2, HGF, PDL-1/2 and IL-6 mRNA were all significantly increased in expression. This contribution indicates the direct ability of the hnRNPA2/B1 protein to influence the immunophenotype of f-LSCs, potentially augmenting their immunosuppressive potential and immunomodulatory action. Although it has not been tested, in our opinion, future genetic engineering of human MSCs might utilize hnRNPA2/B1 gene knockout to potentially enhance the immunosuppressive activity of MSCs.

Finally, to explain whether the anti-inflammatory effects induced by hnRNPA2/B1 were NF-ĸB-dependent, we used parthenolide as pivotal inhibitor of inflammatory responses in Jurkat T cells [45]. The most consistently reported mechanism by which parthenolide inhibits the NF-κB pathway is by directly binding to NF-κB subunits. At a high dose, parthenolide significantly downmodulated the hnRNPA2/B1 mRNA levels as a consequence of the potent anti-inflammatory action evoked by NF-ĸB blocking. These findings describe, for the first time, the hnRNPA2/B1 as a novel inflammatory mediator and target gene of the NF-ĸB pathway. Furthermore, co-immunoprecipitation of hnRNPA2/B1 with the p65/RelA provided new evidence of possible involvement of the hnRNPA2/B1 in regulation of the NF-ĸB transcriptional control. Additional investigations need to be conducted to demonstrate the possible role of the hnRNPA2/B1 protein in the induction of NF-ĸB accumulation and nuclear activation.

This study is limited by its retroprospective design and small cohort of healthy controls and patients. Moreover, at this step in the investigation, we still cannot correlate the high hnRNPA2/B1 expression level to the inflammatory status of AED patients. Nevertheless, a clear relationship between hnRNPA2/B1 downmodulation and its anti-inflammatory effects in autoreactive PBMCs was largely proved at proteomic and transcriptomic levels by different assays. In this light, we speculated a promising attempt to enhance our understanding on the pathophysiology of AEDs and their immunological causes currently monitored through symptomatic therapies.

## 5. Conclusions

In this study, we outline the ways in which f-LSCs sense and control inflammation and immune tolerance induction in PBMCs of AED patients, highlighting the central role of the hnRNPA2/B protein in this process. The novel activation role attributed to this multifunctional protein may eventually favor the induction of inflammatory autoimmune response in etiopathogenesis of the AEDs. Furthermore, the suppression of hnRNPA2/B1 activity could represent an excellent opportunity to develop new pharmacological tools useful in all those conditions, requiring tolerance induction or selective immunosuppression, as in AEDs.

## Figures and Tables

**Figure 1 jcm-09-00009-f001:**
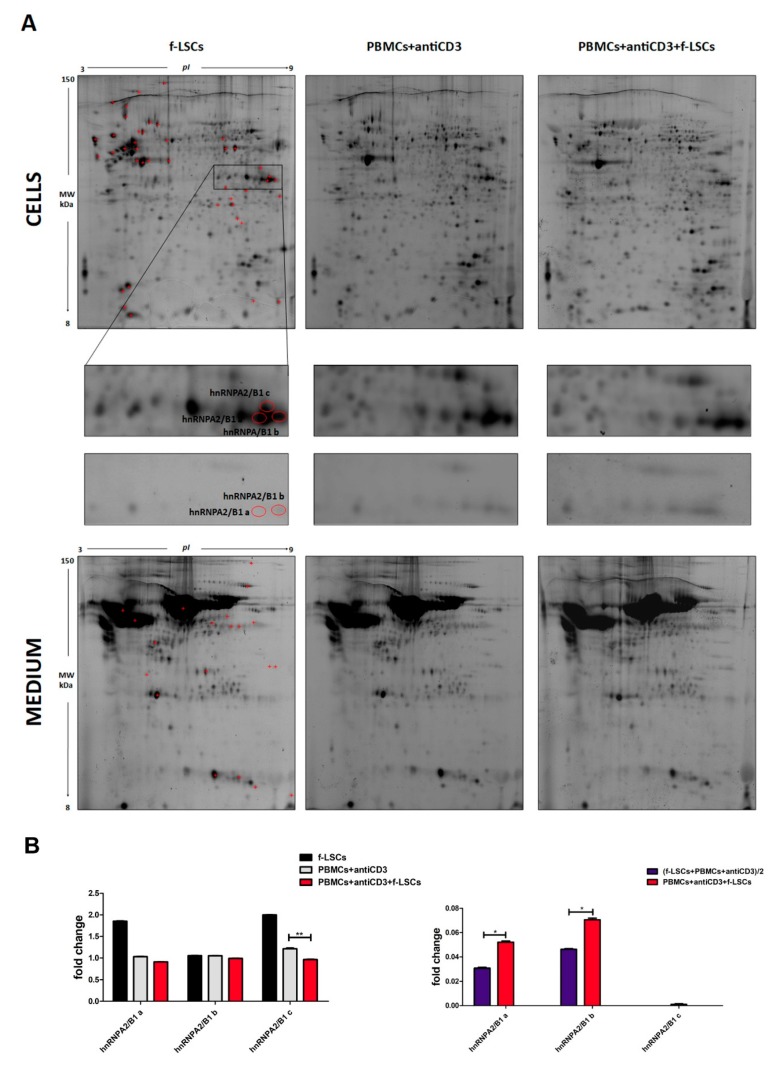
Differential proteomic and secretome analysis by 2D-DIGE and quantification of protein levels. (**A**) Representative miniatures of six 2D silver-stained gels of activated PBMCs, f-LSCs and cocultures are reported. The total protein extracts and the secreted proteins were analysed by 2D-DIGE (upper and lower panel respectively). The graphic red box highlights the protein spots of the three hnRNPA2/B1 isoforms indicated with alphabetical letters. (**B**) The histograms summarize the protein level variations. Intracellular and secreted hnRNPA2/B1 isoforms are plotted as fold change compared to the f-LSC-untreated PBMCs (left panel) or to the sum of the proteins individually released by f-LSCs and PBMCs divided for two (right panel) respectively. Data are presented as means ± SE in each histogram. * *p* < 0.05, ** *p* < 0.02.

**Figure 2 jcm-09-00009-f002:**
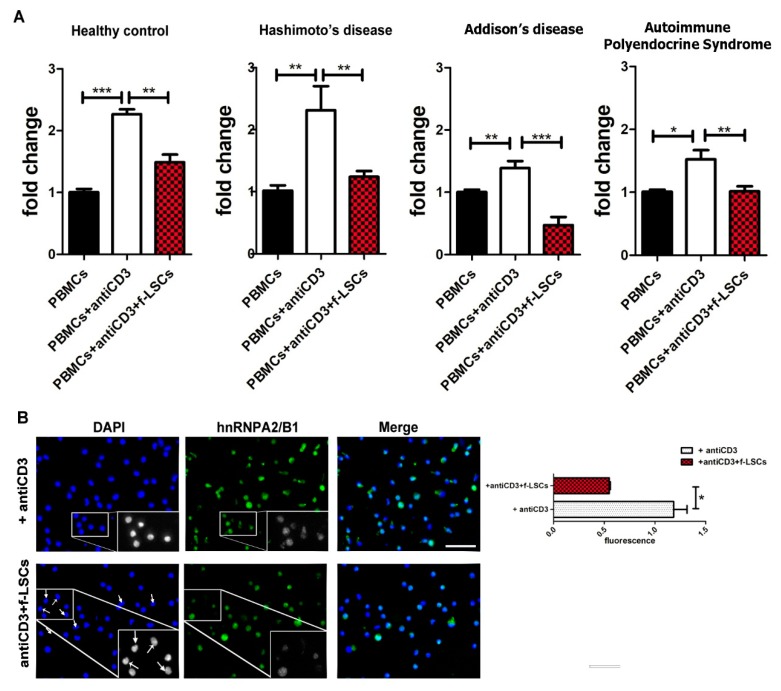
Downmodulation of hnRNPA2/B1 expression in AED patients. (**A**) qRT-PCR shows the hnRNPA2/B1 mRNA levels in activated PBMCs and after 72 h of coculture. Molecular detection of hnRNPA2/B1 expression was assessed in different groups of AED patients (Hashimoto’s, Addison’s disease and APS) compared to healthy controls. Untreated PBMCs were used as negative controls. (**B**) The pictures show immunofluorescence staining for hnRNPA2/B1 protein in activated PBMCs of AED patients before and after f-LSC treatment at 40X objective. The arrows signalize negative nuclei for hnRNPA2/B1 staining in f-LSC-treated PBMCs. Inside the squares, some representative enlarged fields in grayscale are reported. Intensity quantification was calculated by ImageJ software.All pictures are representative of five independent experiments. Data in the histograms are presented as means ± SE with * *p* < 0.05, ** *p* < 0.02, *** *p* < 0.001.1D-western blot. DAPI: 4′,6-diamidino-2-phenylindole.

**Figure 3 jcm-09-00009-f003:**
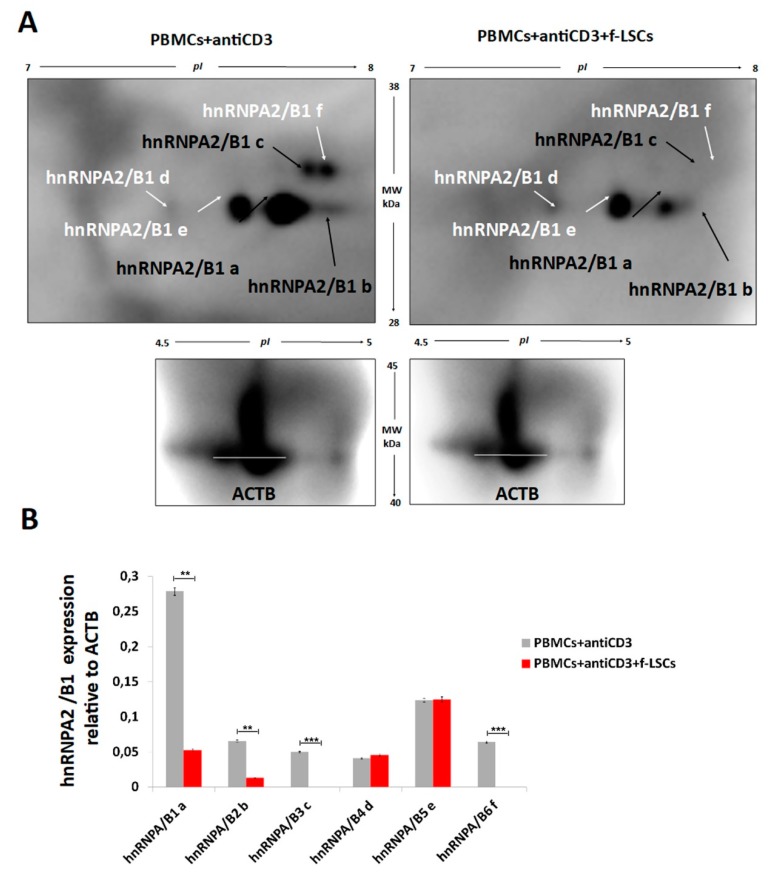
2D-Western blot validations of hnRNPA2/B1 expression in AED patients. (**A**) Proper antibodies (anti-actin and anti-hnRNPA2/B1) were used for bidimensional loading of protein samples harvested from AED patients. The arrows mark all the hnRNPA2/B1 isoforms identified. (**B**) The corresponding quantitative measurement of optical density is reported in the histogram. The images are representative of five experiments. Data are presented as means ± SE with * *p* < 0.05, ** *p* < 0.02, *** *p* < 0.001. ACTB: β actin protein.

**Figure 4 jcm-09-00009-f004:**
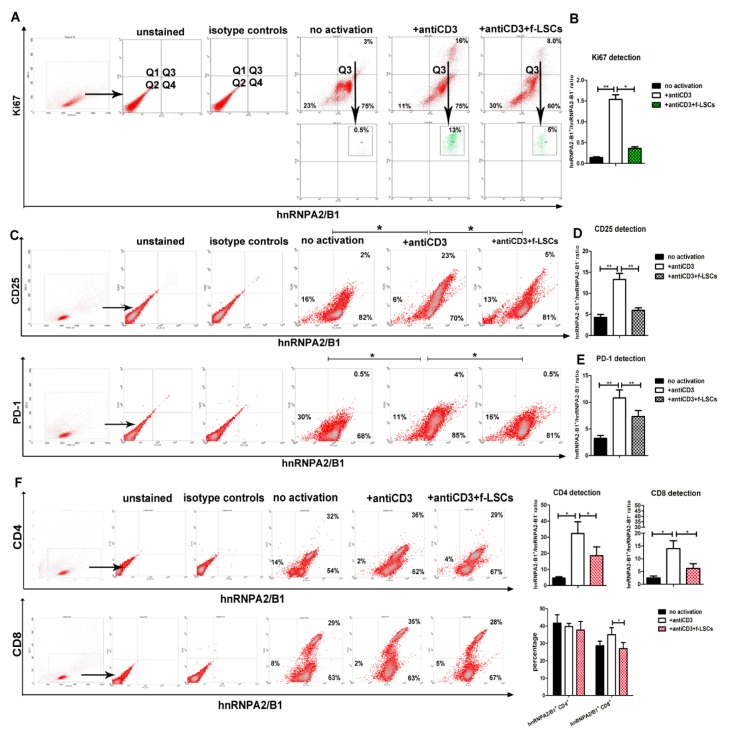
hnRNPA2/B1 downmodulation affects Ki67+, CD25+, PD-1+, CD8^+^ lymphocytes of AED patients. (**A**) Representative dot plots of activated PBMCs collected by AED patients and stained for Ki67 and hnRNPA2/B1. In the upper right quadrant (Q3) of the dot plots, the gating strategy is shown. (**B**) In histogram, the dual staining Ki67-hnRNPA2/B1 of AED PBMCs, reported as hnRNPA2/B1^+^:hnRNPA2/B1^−^ ratio is shown. (**C**) A comparison of FASC analysis for the activation markers CD25, PD-1 and hnRNPA2/B1 is reported. (**D**) and (**E**) In histograms, the dual staining CD25-hnRNPA2/B1 and PD-1-hnRNPA2/B1 of AED PBMCs, reported as hnRNPA2/B1^+^:hnRNPA2/B1^−^ ratios, are respectively shown. (**F**) Representative dot plot shows the percentages of hnRNP A2/B1 coexpression with the CD4^+^ or CD8^+^ T cell markers (left panels). The numbers in the corner of quadrants correspond to the percentage of related cell population inside any quadrant. The same data are reported in a percentage graph (lower-right panel) or calculated as hnRNPA2/B1^+^:hnRNPA2/B1 ratio in each condition during the CD4 and the CD8 detection (upper-right panel). The data show the averaged results from five independent experiments and are presented as means ± SE in each histogram. * *p* < 0.05, ** *p* < 0.02, *** *p* < 0.001.

**Figure 5 jcm-09-00009-f005:**
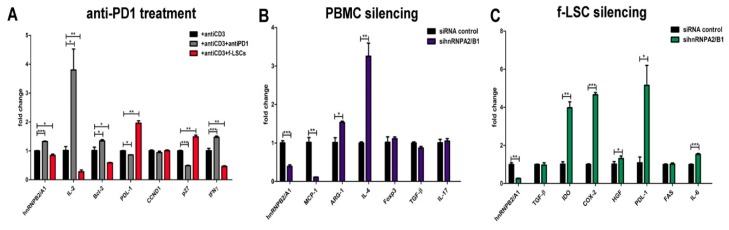
Immunofunction of hnRNPA2/B1 in PBMCs of AED patients and in f-LSCs. (**A**) PBMCs collected from AED patients were activated with anti-CD3 mAb and incubated with anti-PD-1 human recombinant protein or f-LSCs. After anti-PD-1 blocking, the molecular detection of IL-2, Bcl-2, PDL-1, p27Kit, CCND1and IFN-γ mRNA by qRT-PCR was assessed. (**B**) hnRNPA2/B1 siRNA knowdown in activated PBMCs of AED patients is shown. At 72 h post electroporation, expression for FOXP3, TGF-β, IL-17A, Arg-1 and IL-4 was detected. (**C**) Expression levels of several immunosuppressive and tolerogenic markers (TGF-β, IDO, COX-2, HGF, PDL-1/2, FAS, IL-6) in f-LSCs after chemical transient silencing of hnRNPA2/B1 protein are reported. All results are shown as mean ± SD of five independent experiments, * *p* < 0.05, ** *p* < 0.02, *** *p* < 0.001.

**Figure 6 jcm-09-00009-f006:**
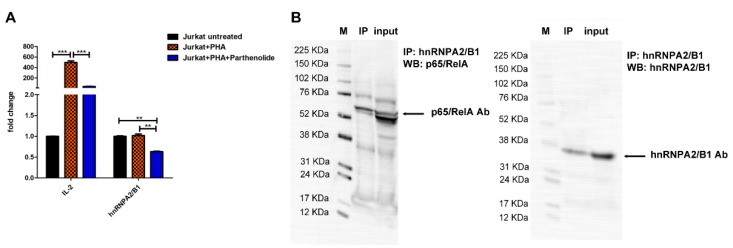
hnRNPA2/B1 mRNA levels are influenced by NF-ĸB transcription factor activity in Jurkat T cells. (**A**) Jurkat cells were activated for 5 hours with 10 µg of PHA and treated with 10 µM of parthenolide to induce or block the NF-kB nuclear action respectively. Unstimulated cells were used as negative controls. All results are shown as mean ± SD of five independent experiments, ** *p* < 0.02, *** *p* < 0.001. (**B**) Protein extracts prepared from Jurkat cells were immunoprecipitated with human anti-hnRNPA2/B1 antibody. The immune complexes and the input (50 µg of total extracts used in the immunoprecipitation) were analysed by immunoblotting with antibodies specific to human p65-RelA and hnRNPA2/B1. M: marker; IP: immunoprecipitated.

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
