# Peer review of "Anti-Inflammatory Action of Heterogeneous Nuclear Ribonucleoprotein A2/B1 in Patients with Autoimmune Endocrine Disorders"

_jcm, 2019, doi:10.3390/jcm9010009_

Round 1

Reviewer 1 Report

line 49: Hahsimto’s thyroidits instead of Graves’ disease will be more appropriate, because Graves’ disease is essentially hyperplasia of thyrocytes but not destruction. line 145-146: the sentence should be reformatted. line 201-202, line 589; should be reformatted properly. line 309: the title of legend for Figure 1 is incomplete. The reviewer cannot see the panel of Fig. 6, although the legend can be seen. The title of Table T1 should be written correctly.

Author Response

Response to Reviewer 1 Comments

Point 1: line 49: Hahsimto’s thyroidits instead of Graves’ disease will be more appropriate, because Graves’ disease is essentially hyperplasia of thyrocytes but not destruction.

Response 1: we removed from the list of autoimmune cytodestruction disorders the Graves’ disease as you suggested.

Point 2: line 145-146: the sentence should be reformatted.

Response 2: thanks for your suggestions. We reformatted the sentences.

Point 3: line 201-202, line 589; should be reformatted properly.

Response 3: we properly reformatted them as you suggested.

Point 4: line 309: the title of legend for Figure 1 is incomplete.

Response 4: we extended the title of legend for Figure 1 at line 269 as you suggested.

Point 5: The reviewer cannot see the panel of Fig. 6, although the legend can be seen.

Response 5: Sorry for the inconvenience. We inserted the Figure 6 above the legend.

Point 6: The title of Table T1 should be written correctly.

Response 6: we replaced the title for Table S1 correctly.

Reviewer 2 Report

The authors here present a study in which they expore how fibroblast-limbal stem cells (f-LSCs) may influence the PBMC properies/balance in patients with endocrine disorders. The study is novel, and interesting, but the main message disappears in all the text, and it seems unfocused.

I have some concerns

Major concerns:

The introduction starts with the results (I think), and this is a very unusual and suboptimal way of leading the reader to understand WHY the authors are doing this. In my view, the intro should start with the lack of knowledge on pathogenic mechanisms in autoimmunity, not wth the results. WHAT is the hypothesis here? (I think I know, but the message is kept within the lines and not clear Please focus the whole intro The methods are described good The results are also unfocused, but if the intro is more "direct", I think the result part will follow and it will become easier to read. The discussion part is better. Here, I find clues for the hypothesis. But please do not repeat the intro to the discussion. The table and figures are easy to catch and therfore good

Minor concerns:

  1. Please change "PAS" to "APS", which I think is much more used

  2. There are much more females than males in the disease groups and vice versa in the control group. Could this interfer with the results?

  3. Does 12 h fasting also mean "fasting from medications"? Could this interfer with results; e.g for Addison's diease: Glucocorticoids have a large effect on immunomodulatory molecules. Using patient materials for individuals that have no glucocorticoids in the morning could be a suboptimal design?

4. I guess the first Suppl Table nr 2 is actually number 1.

5. It is not common to think that all autoimmune endocrine disorders have exactly the same pathogenesis.....here you mix different both monogenic and polygenic forms. I would kind of like an argument for that.

Author Response

Response to Reviewer 2 Comments

The authors here present a study in which they explore how fibroblast-limbal stem cells (f-LSCs) may influence the PBMC properies/balance in patients with endocrine disorders. The study is novel, and interesting, but the main message disappears in all the text, and it seems unfocused.

I have some concerns

Major concerns:

Point 1: the introduction starts with the results (I think), and this is a very unusual and suboptimal way of leading the reader to understand WHY the authors are doing this. In my view, the intro should start with the lack of knowledge on pathogenic mechanisms in autoimmunity, not wth the results. WHAT is the hypothesis here? (I think I know, but the message is kept within the lines and not clear Please focus the whole intro The methods are described good The results are also unfocused, but if the intro is more "direct", I think the result part will follow and it will become easier to read. The discussion part is better. Here, I find clues for the hypothesis. But please do not repeat the intro to the discussion. The table and figures are easy to catch and therfore good.

Response 1: we greatly appreciated your suggestions and rightful concerns. As you requested we modified the introduction section focusing on the principal scope of our study according your suggestions. Furthermore, in the discussion section we eliminated the repetitive sentences/concepts.

Minor concerns:

Point 2: Please change "PAS" to "APS", which I think is much more used

Response 2: we changed "PAS" to "APS" as you suggested.

Point 3: there are much more females than males in the disease groups and vice versa in the control group. Could this interfer with the results?

Response 3: unfortunately this inconvenience could be a limitation due to the difficulty to select male patients with Hashimoto’s thyroiditis. In any case, in this first step of investigation, we planed a retrospective study to identify novel markers of disease in AED patients and explaining their possible role/function and potentiality.

Point 4: Does 12 h fasting also mean "fasting from medications"? Could this interfer with results; e.g for Addison's diease: Glucocorticoids have a large effect on immunomodulatory molecules. Using patient materials for individuals that have no glucocorticoids in the morning could be a suboptimal design?

Response 4: each blood sample was drawn after 12h of fasting from foods. All selected patients, included PAI patients, were on stable hormone replacement therapy before entering the study and they had duration of disease of at least five years. Furthermore, Glucocorticoids were administrated at a dosage able to restore the physiologic blood levels of cortisol.

Point 5: I guess the first Suppl Table nr 2 is actually number 1.

Response 5: your comment is correct. We replaced the title for Supplemetal Table S1 as you suggested.

Point 6: It is not common to think that all autoimmune endocrine disorders have exactly the same pathogenesis.....here you mix different both monogenic and polygenic forms. I would kind of like an argument for that.

Response 6: A subgroup analysis for different set of patients was carried out by use qRT-PCR analysis for hnRNPA2/B1 mRNA detection as shown in Figure 2A. No significant differences were found in the three groups of patients with a downmodulation level of the selected protein similar in each group. For this reason the experiments that followed referred to AED patients.